# Do-It-Yourself digital archaeology: Introduction and practical applications of photography and photogrammetry for the 2D and 3D representation of small objects and artefacts

**Jacopo Niccolò Cerasoni**[1,2]* , **Felipe do Nascimento Rodrigues**[3], **Yu Tang**[4], **Emily Yuko Hallett**[1]

**1** Pan-African Evolution Research Group, Max Planck Institute for the Science of Human History, Jena, Germany, **2** Institute of Archaeological Sciences, Eberhard Karls University Tübingen, Tübingen, Germany, **3** Department of Archaeology, Centre for the Archaeology of the Americas, Material Lives -Transdisciplinary Research Group, University of Exeter, Exeter, United Kingdom, **4** Cygames, Inc., Shibuya-ku, Tokyo, Japan

☯ These authors contributed equally to this work.
* cerasoni@shh.mpg.de

**Data Availability Statement:** The article does not contain data and the data availability policy is not applicable to the article.

## Abstract

Photography and photogrammetry have recently become among the most widespread and preferred visualisation methods for the representation of small objects and artefacts. People want to see the past, not only know about it; and the ability to visualise objects into virtually realistic representations is fundamental for researchers, students and educators. Here, we present two new methods, the 'Small Object and Artefact Photography' ('SOAP') and the 'High Resolution "DIY" Photogrammetry' ('HRP') protocols. The 'SOAP' protocol involves the photographic application of modern digital techniques for the representation of any small object. The 'HRP' protocol involves the photographic capturing, digital reconstruction and three-dimensional representation of small objects. These protocols follow optimised step-by-step explanations for the production of high-resolution two- and three-dimensional object imaging, achievable with minimal practice and access to basic equipment and softwares. These methods were developed to allow anyone to easily and inexpensively produce high-quality images and models for any use, from simple graphic visualisations to complex analytical, statistical and spatial analyses.

## Introduction

Archaeologists continuously apply novel approaches from complementary disciplines for the better understanding of archaeological contexts and past human activity and behaviour, both in the field and in the laboratory. For example, archaeological observations of stratigraphies (i.e. the study of accumulation of sediments and materials through time) first originated from geology in the 15th century [1], and modern archaeological scientific methods such as

**Funding:** J.N.C. and E.Y.H are funded by the Pan-African Evolution Research Group, based at the Max Planck Institute for the Science of Human History, Jena, Germany. F.N.R. is funded by the University of Exeter College of Humanities International PhD Studentship. Y.T. is funded by Cygames, Incorporated based in Tokyo, Japan. The funders had no role in study design, data collection and analysis, decision to publish, or preparation of the manuscript.

**Competing interests:** The authors have declared that no competing interests exist.

biomolecular and chemical archaeology apply theory and method from the disciplines of biology, genetics and chemistry. Similarly, the study of archaeological material culture and the methods used for the visual representation of artefacts has developed at an incredibly fast rate following the innovations in digital technology that have occurred during the past decades.

The ability to visually represent archaeological materials has always been a fundamental part of archaeological publications and dissemination, as the study of material culture (e.g. stone tools, pottery, metal objects, organic materials, etc.) is one of the principal factors of archaeological research. Traditional means of visual representation of material culture commonly include illustrations in physical or digital formats [2]. However, in the recent decades, and following an expansion of accessibility to digital equipment, photographic and three-dimensional (3D) representations of material culture have become dominant methods in the field.

Photography is undoubtedly the most common medium in use today to represent artefacts in archaeological research [3]. Its origins in this field can be traced back to the mid-nineteenth century [4], and the use of this photography for the recording archaeological artefacts is nearly as old as photographic technology [5]. Photography is generally used to record artefacts, archaeological sites, landscapes, and monuments [3,5,6].

A major improvement in the visual representation of archaeological artefacts has been the shift from an illustrative and artistic photographic style, to a more analytical and objective format. While artistic photographic styles are still employed in cases where public outreach and science communication are of primary interest, analytical and objective photography has become essential for the proper representation of archaeological materials and contexts following proper scientific methods. Examples of this are the increased use of satellite imagery as a primary data source for archaeological surveying [7] and high-resolution microscopic photography as an essential method for the identification and recording of past tool uses [8].

Thanks to its inherent value in the visual communication of past human behaviour, archaeological photography has been keenly scrutinised and ultimately improved over time. Major improvements to archaeological photography include methodological and technical improvements in the form of development of photographic equipment and digital control of photographic products and environments [4], the critical theoretical evaluation of the objective nature of photography [9], and the contemporary practical and theoretical reassessment of the relationship between archaeology and photography [10]. Overall, these advancements have raised photography to a status beyond that of an illustrative medium. Nevertheless, gaps remain in the practical teaching of archaeological photography, as can be seen in the lack of university-level Archaeology programmes offering photographic training [3]. Several valuable resources exist for archaeologists that discuss the practical aspects of photography, such as the BAJR guides introducing photography [11], and detailed site and artefact photography manuals [12,13].

Here we present the "Small Object and Artefact Photography", or 'SOAP', protocol as an addition to the field of archaeological photography. This new protocol combines a detailed, concise, and user-friendly workflow that covers the entire photographic acquisition and processing process, thereby contributing to the replicability and reproducibility of high-quality photographs. By clearly explaining every step of the process, and adding theoretical and practical notions to steps explaining camera technical functionalities, the 'SOAP' method shows users how to take high quality photographs and also described the reasons why photographs can be successful or unsuccessful.

Photogrammetry, like photography, has advanced as a method in archaeology over the last decade, resulting in a significant increase in use over the last six years [14,15]. Its growth in popularity among researchers, heritage professionals, and the public is mostly due to its exceptional ability to bring people even closer to objects and landscapes, in combination with its low

cost in comparison to other 3D recording methods (e.g. structured light, laser, CT scanning, and terrestrial/aerial LiDAR) [2,14]. Photogrammetry has been used in a range of archaeological contexts, including faunal and paleontological studies [16], lithic use wear analysis [17], small artefact analysis [18], and site photogrammetric surveys [19].

Following the modern computational revolution in archaeology [20], photogrammetry finds itself at the spotlight of archaeological visual representation, with continuous technical and methodological developments [18,21]. While data visualisation methods are expanding at an exponential rate, the use of photogrammetric methods for new analytical techniques and analyses is yet to be fully explored; this issue has been raised by other researchers [15–20]. To address this, we present here the protocol for the High Resolution "DIY" Photogrammetry ('HRP') Method. This new protocol makes photogrammetry more accessible and less time consuming for beginners by providing a detailed workflow of each step and streamlining the entire photogrammetry process. Our protocol covers all stages of photogrammetry—from image acquisition to post processing—and allows 'more time for focusing on photogrammetry's analytical applications. Our aim in streamlining photogrammetry and making it widely accessible is to allow archaeologists to further integrate photogrammetry in archaeological research.

The 'SOAP' and 'HRP' protocols offer clear step-by-step processes that anyone can learn and put into practice. However, it is important to note that a good photograph or a good three-dimensional model will always be just a visual representation of the visualised artefact. For this reason, it is important to note that a good understanding of the artefact's morphology, technological characteristics and context will always be necessary for the correct interpretation of the visualised material culture. Furthermore, both methods will inevitably encounter limitations depending on the used equipment, workflow variations, and subjective evaluations during their application. Quantitative methods of image analysis [22–24] were not applied in our protocols as they fall outside the scope of the applied methods in archaeology that we present here. Both in terms of photographic and photogrammetric documentation, minor differences in image quality will occur depending on a range of variables that will be person- and case-specific. Improving equipment capabilities both in terms of hardware and software functionality will likely automatically result in better and more efficient final products. Increased time spent on practicing the presented methods will also exponentially improve their application and outcomes.

The 'SOAP' and 'HRP' protocols were developed using Adobe Camera Raw ©, Adobe Photoshop 2021 ©, RawDigger ©, DxO Photolab ©, and RealityCapture ©, as they have native functions and tools that make them easier and faster compared to other comparable softwares. Although most of the used softwares in the 'SOAP' and 'HRP' protocols are readily available in academic environments, these methods can be applied to any other non-subscription based softwares with similar features. In this regard, free and/or open-access softwares can be readily used, albeit with minor changes in the application of some of the presented steps depending on the used software's functionalities. For raster-based softwares used for both photography and photogrammetry, Adobe Photoshop © can be used, while free to access software such as GIMP © and Krita ©,or single-purchase products such as Affinity Photo © can be used. For 3D Reconstruction photogrammetric softwares, such as RealityCapture ©, a good free and open-source alternative is Meshroom ©.

## Materials and methods

The protocols described in this peer-reviewed article are published on protocols.io, https://dx.doi.org/10.17504/protocols.io.b53zq8p6 ('SOAP' Protocol) and https://dx.doi.org/10.17504/protocols.io.b53xq8pn ('HRP' Protocol), and are included for printing supporting information file 1 and 2 with this article.

## Expected results

While a variety of publications on artefact photography and small object photogrammetry already exist, with the application of the 'SOAP' and 'HRP' methods it is expected that users will be able to produce high-quality and publishable two- and three-dimensional visualisations of their archaeological artefacts independently and without the necessary dependency of other methodological sources. Furthermore, with enough practice over time and access to the softwares listed above, anyone who is interested in archaeological material culture, whether for personal, educational, or professional reasons, will be able to do so while keeping time and costs as efficient and low as possible.

Of particular importance, with the application of the 'HRP' Method, differentiations in skill or experience level will result in little to no difference in the application and comprehensibility of the method. Anyone will be able to produce high quality 3D scans at a fraction of the price of other scanning techniques, such as light structured scanning, laser scanning, or CT-scanning. The application of this method makes high-resolution models achievable using beginner or intermediate level equipment and at a much higher resolution compared to other expensive scanning methods.

Overall, whether for simple visualisation or more complex analytical purposes, the protocols presented here will offer the possibility to produce high quality visualisations of artefacts. It is therefore expected that any users of these protocols can produce photographs and photogrammetric models for: (1) academic and general audience publication, (2) quantitative and analytical purposes (e.g. geospatial, statistical, morphological, functional), or (3) public outreach (e.g. printable 3D models, museums, exhibitions, children's activities).

## Supporting information

**S1 Fig.**
(PNG)

**S1 File. Small Object and Artefact Photography—'SOAP' protocol.** also available on protocols.io.
(PDF)

**S2 File. High Resolution "DIY" Photogrammetry—'HRP' Protocol.** also available on protocols.io.
(PDF)

## Acknowledgments

We would like to thank the Pan-African Evolution Research Group and the Max Planck Institute for the Science of Human History for their support during the development of this study. We sincerely thank the reviewers of this study, particularly Dr. Samantha T. Porter (University of Minnesota) for her extremely detailed and in-depth review. E.Y.H. thanks Curtis W. Marean (Arizona State University) for assistance with best practices in macro photography. The following study was carried out as part of the Archaeological Creative Tank.

## Author Contributions

**Conceptualization:** Jacopo Niccolò Cerasoni.

**Investigation:** Jacopo Niccolò Cerasoni, Felipe do Nascimento Rodrigues, Yu Tang, Emily Yuko Hallett.

**Methodology:** Jacopo Niccolò Cerasoni, Felipe do Nascimento Rodrigues, Yu Tang, Emily Yuko Hallett.

**Project administration:** Jacopo Niccolò Cerasoni.

**Supervision:** Jacopo Niccolò Cerasoni.

**Validation:** Jacopo Niccolò Cerasoni, Felipe do Nascimento Rodrigues, Yu Tang, Emily Yuko Hallett.

**Writing – original draft:** Jacopo Niccolò Cerasoni, Felipe do Nascimento Rodrigues.

**Writing – review & editing:** Jacopo Niccolò Cerasoni, Felipe do Nascimento Rodrigues, Yu Tang, Emily Yuko Hallett.

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
