## [Decision Letter · Decision Letter 0]

26 Jan 2022

PONE-D-21-37779Do-It-Yourself Digital Archaeology: Introduction and Practical Applications of Photography and Photogrammetry for the 2D and 3D Representation of Small Objects and ArtefactsPLOS ONE

Dear Dr. Cerasoni,

Thank you for submitting your manuscript to PLOS ONE. After careful consideration, we feel that it has merit but does not fully meet PLOS ONE’s publication criteria as it currently stands. Therefore, we invite you to submit a revised version of the manuscript that addresses the points raised during the review process.

We look forward to receiving your revised manuscript.

Kind regards,

Radu Iovita

Academic Editor

PLOS ONE

Journal Requirements:

2. Please consider modifying the title and Introduction/Discussion section to ensure that you have specified the scientific or medical rationale for this sperm-based sex selection technique.

4. Thank you for providing the following Protocols.io DOI in your submission form [Protocols.io DOI]. In keeping with our submission requirements, please add the Protocols.io DOI to the Methods section of your manuscript as well using this format: “The protocol described in this peer-reviewed article is published on protocols.io, https://dx.doi.org/10.17504/protocols.io[........] and is included for printing as supporting information file 1 with this article.” For more information, please see our submission guidelines: https://journals.plos.org/plosone/s/submission-guidelines#loc-guidelines-for-specific-study-types

Reviewers' comments:

Reviewer's Responses to Questions

**Comments to the Author**

1. Does the manuscript report a protocol which is of utility to the research community and adds value to the published literature?

Reviewer #1: Yes

Reviewer #2: Yes

2. Has the protocol been described in sufficient detail?

Descriptions of methods and reagents contained in the step-by-step protocol should be reported in sufficient detail for another researcher to reproduce all experiments and analyses. The protocol should describe the appropriate controls, sample sizes and replication needed to ensure that the data are robust and reproducible.

Reviewer #1: No

Reviewer #2: Partly

3. Does the protocol describe a validated method?

Reviewer #1: No

Reviewer #2: Yes

4. If the manuscript contains new data, have the authors made this data fully available?

Reviewer #1: No

Reviewer #2: N/A

**5. Is the article presented in an intelligible fashion and written in standard English?**

Reviewer #1: Yes

Reviewer #2: Yes

6. Review Comments to the Author

Reviewer #1: The text in my opinion presents very broadly practice guidelines potentially useful for archaeologist just making their first steps into artifact photography or 3d modeling of archaeological artifacts. I do agree with the basic premise of the presented text, yet I do not believe it is in fact a scientific article nor is it a detailed description of a workflow.

The paper combines authors' view on both object photography and photogrammetry 3d modeling of artifacts. Both these areas are huge and diverse fields in their own rights. Both can be treated as separate disciplines of art, craft and science and each can be studied full time to the MA/M.Sc. level and beyond on various universities around the world at readily available courses.

To attempt to sum up this amount of knowledge into one paper is a difficult task. Possibly bound to present summaries simplified beyond the level which can be supported by data. Thus, the paper presents no data.

In my opinion, the text should be rewritten as at least two separate ones where authors could focus separately on each subject.

Firstly comes the premise of Small Object and Artifact Photography -"SOAP" workflow. Most statements which have found its way into the attached protocol should be developed in concise form into the body of the article and the protocol itself should present actual data, preferably a statistical sample.

The protocol should also explain actual gains of at least one step of the process which authors believe they have perfected beyond what is available now in practice of professional photographers, in on-line sources or in literature. Possibly focusing on one aspect specifically would also help the authors to explain what in their opinion is the decisive quality of a “good” or “bad” photograph.

The issues of focus stacking, illumination control, diffraction, lens selection, background choice or background removal etc., each could warrant a discussion and possibly improvement. In its current form, the paper in this part and attached protocol do not convince that following presented advice to the letter will result in any specific improvement. Most statements are generally “common knowledge” for a working photographer and are only new to an archaeologist venturing first into serious photography.

This part of work in essence will benefit from restructuring. It is vital to show which part of photographic documentation process can be improved. Also, it is important to introduce a quantitative method to show the actual improvement and dataset (large, statistically significant set of photographs) pointing the areas of improvement.

Secondly, part of the text referring to photogrammetry ”HRP” protocol also presents no actual data. Authors did not explain sufficiently in paper nor in protocol what are the markers to measure the quality of a photogrammetric 3d model and how using their specific workflow will make a significant or even measurable difference over just following other workflows, including those proposed by 3d photogrammetry software manufacturers according to this measured parameter.

This parameter has to be explained in order for the research to ever be replicated. The improvement of the parameter should also be presented on a statistical sample of processed 3d models. To create such a body of data would be time-consuming, but would give the paper and proposed protocol some credibility.

As it is now the paper in this part doesn’t present any information and the attached “HRP” protocol is an instructional material no better or worse than any other similar workflow and its usability will only be subjective and dependent on the experience of those who might have used it for their first work in 3d photogrammetry.

Reviewer #2: More public, citable technical guides in archaeology are desperately needed, especially for newer techniques like photogrammetry, and I thanks and commend the authors for their contribution. From a high level, the article does a good job of setting the stage for the two associated protocols. These protocols are well thought out, have many good illustrations, and include lots of important details, but could be reorganized and edited to add more context and be easier to follow for readers who may not be as familiar with photography or 3D scanning. Due to the fact that there are currently not many formal courses on these subjects as mentioned by the authors, it is possible that a significant number of readers will fall into this category. If it has not been done already, having inexperienced users ‘play test’ the protocols could reveal additional ways the protocols could be made more broadly helpful.

Specific comments on the different submitted elements follow.

—--

Article Text:

In the abstract, it may be best to specify that photogrammetry refers to 3D reconstruction / “structure from motion” as technically “photogrammetry” could more broadly refer to the general concept of taking measurements from photographs.

What the authors mean by ‘illustrative’ versus ‘object’ representations of objects in the fourth paragraph of the second page is not exactly clear and could be better defined. What in the authors’ view are the goals of each type of representation?

In the third paragraph on page 4, I would add LiDAR (perhaps both terrestrial and aerial) to the list of 3D recording methods. It may be more appropriate to use e.g. as opposed to i.e. This is also true perhaps of the list of types of material culture in the second paragraph of the introduction.

As many of the software applications described in the protocols are not free (Adobe Photoshop, Reality Capture, and ArcGIS for example) it is maybe not appropriate to highlight free software in the expected results section. The authors could state that much of the software is likely already accessible to folks in an academic environment, or that their photography protocols could be used with free alternatives to the software applications described. It’s also quite minor, but in my experience practice over time does definitely improve results for both artifact photography and photogrammetry, so

—--

‘SOAP’ Protocol:

It may be worth mentioning copy stands as a mounting option, as in my experience many academic departments have these laying around and they can be more stable than many models of tripod.

For readers without much experience, it may be worth defining a macro lens and stating why it is most suited to this type of photography. They could also refer readers to section 8 (Step 2).

The authors could also suggest that readers use a two or ten second timer on their camera if remote control is not possible. This can minimize blurriness due to shake. Tethering to a computer is another option. The reasons why this should be done should be stated upfront (Step 2.2).

Satin paper and velvet are mentioned, but it is not explained why these materials are good options. It would be good to add an explanation (e.g. they are not reflective) (Step 3.1).

There is a missing word in the explanation of the RAW format (...preferred as they are the least processing…” The description of RAW and TIFF files as being more ‘archival’ could be helpful. Inexperienced users may not understand what compression is and how it can degrade an image. I would suggest adding an explanation. (Step 6)

I would move a discussion of the “exposure triangle” earlier in section 8 to give context to the suggested camera settings (aperture, shutter speed, and ISO). Some sort of visualization of this concept would be helpful.

I think it would make sense to discuss ISO before the shutter speed, as it should usually be set at a low value, while exposure time is more likely to be variable depending on things like material properties, ambient light in the environment, etc.

This is not going to affect most users, but some lenses do not have both a focus and zoom ring and focus must be adjusted by moving closer / further to the subject. For example our lab uses a Laowa 25mm f/2.8 2.5-5X Ultra-Macro Lens for very up close photography / photography of very small objects. Going into this in detail is likely not worth it, but it may be helpful to mention that the example depicted applies to most but not all lenses (Step 10).

A visual of what ‘exposure’ means on camera and what it does would be helpful, as shutter speed is often colloquially referred to as exposure (Step 12)

An explanation of white balance would be helpful during the photography instructions (and not just the editing instructions), especially if people do not end up shooting RAW.

Some consider non-linear adjustments like curves and levels to be non-ideal for scientific publishing as they affect certain pixels more than others. See for example Rossner and Yamada 2004 (https://doi.org/10.1083/jcb.200406019). Photo filters are a linear alternative, but must be done by eye. An alternative would be to have users disclose any manipulation done on the image in publication (Step 18).

It could be helpful to suggest that readers also save their work as a PSD file if it needs additional manipulation later on (for example to be able to copy the cut out object and use it in a collage later on) (Step 27).

—--

‘HRP’ Protocol:

A list of software in addition to hardware at the start of the protocol would make the article easier to read.

It would be helpful if the authors explained why using a cell phone is not as accurate and precise (e.g. lack of manual control) (End of Step 1).

It would be helpful to explain why a solid color background is ideal (e.g. it helps separate the target object from the background). I am not familiar with Reality Capture, but for Agisoft Metashape for example it allows for easier masking. It also may be helpful to describe the importance of contrast between the object and background as was done in the ‘SOAP’ protocol (Step 2).

As mentioned in my comments on the ‘SOAP’ protocol, in order to reduce blurriness from motion, setting the camera on a two second timer is an alternative if a remote shutter is not a possibility (Step 3).

Photography terms such as aperture are not as well defined in this protocol as they are in the other. Perhaps the reference to ‘SOAP’ could be moved earlier.

It is my understanding that for many lenses diffraction begins at f/11, and that thus f/16 should be avoided. See for example Verhoeven 2016 (Basics of Photography for Cultural Heritage Imaging, https://biblio.ugent.be/publication/8050621/file/8050622.pdf).

An explanation that high ISO can result in noise being added to an image may be helpful here. A sample image would be particularly welcome.

There is a typo where Rawdigger is spelled Rawfigger. (Step 8).

There is a typo at the start of Step 11 (I believe “repeat rolling” should be “repeat rotating”). A distinction perhaps between rotating the turntable and turning the object upside down should be made, as it is currently confusing. Diagrams of the different photography options (i.e. turning the object around vs moving the camera) would be very helpful.

For scientific purposes, in my opinion a scale should be necessary and not just recommended (Step 11.3).

I would suggest adding that objects should be rotated 20 degrees between images as expressing this as 18 times per rotation isn’t necessarily intuitive (Step 12).

Is there a reason DxO PhotoLab is used for RAW imaging processing in this workflow, while Adobe Photoshop is used in the other workflow? I do like the suggestion of LibRaw as a free alternative, and wonder if it is worth mentioning that in the SOAP protocol? In both cases, if readers do not have access to software to edit RAW files it may be worth stating that they can still achieve good results (Step 14).

The adjustments in steps 16 and 17 seem like they could be useful for photogrammetric processes, but are non-linear and thus not ideal for scientific use (see again Rossner and Yamada 2004 - https://doi.org/10.1083/jcb.200406019). It may be worth being explicit about the need to disclose this workflow when publishing models created using this protocol.

Since ‘Metashape’ is a newer name for the software, I would refer to it by its full name of ‘Agisoft Metashape.’ Including some advantages and disadvantages of the applications mentioned would be welcome (Step 14).

The protocol should define ‘too many faces’ for inexperienced users. This could include an explanation that a high number of polygons can be difficult to load and manipulate on standard computers for example (Step 29). A suggestion for an upper limit on the number of polygons could help readers.

New users may not know what a texture map is, so an explanation and even an image could be added to make this clearer (Step 30.1).

An explanation of different 3D file formats would be welcome in the export section, as was done for 2D images in the ‘SOAP’ protocol.

The inclusion of the video renders and interactive 3D models is great! A discussion of different options for sharing 3D models would be welcome for newer users, but I think not absolutely necessary to this particular protocol.

For scientific purposes, users will need to record metadata about their photography and processing steps (e.g. number of photos used, software settings, etc.) . This should be mentioned somewhere in the protocol.

Finally, if possible I would encourage the authors to include a sample dataset of RAW or pre-processed images so users can experiment with the Reality Capture portion of the protocol without having to capture their own images. This would be particularly useful for example as a teaching resource for courses on digital archaeology.

7. PLOS authors have the option to publish the peer review history of their article (what does this mean?). If published, this will include your full peer review and any attached files.

Reviewer #1: No

Reviewer #2: **Yes: **Samantha T. Porter

---

## [Author Response · Author response to Decision Letter 0]

28 Mar 2022

Given the extensive length of our responses to reviewers, all the the responses, comments and manuscript edits can be viewed in the attached file "Response to Reviewers Guide".

---

## [Editor Report · Decision Letter 1]

4 Apr 2022

Do-It-Yourself Digital Archaeology: Introduction and Practical Applications of Photography and Photogrammetry for the 2D and 3D Representation of Small Objects and Artefacts

PONE-D-21-37779R1

Dear Dr. Cerasoni,

We’re pleased to inform you that your manuscript has been judged scientifically suitable for publication and will be formally accepted for publication once it meets all outstanding technical requirements.

Kind regards,

Radu Iovita

Academic Editor

PLOS ONE
---

## [Editor Report · Acceptance letter]

6 Apr 2022

PONE-D-21-37779R1 

Do-It-Yourself Digital Archaeology: Introduction and Practical Applications of Photography and Photogrammetry for the 2D and 3D Representation of Small Objects and Artefacts 

Dear Dr. Cerasoni:

I'm pleased to inform you that your manuscript has been deemed suitable for publication in PLOS ONE. Congratulations! Your manuscript is now with our production department. 

Kind regards, 

on behalf of

Dr. Radu Iovita 

Academic Editor

PLOS ONE